EMBO
Molecular Medicine

# Chlorambucil targets BRCA1/2-deficient tumours and counteracts PARP inhibitor resistance

Eliana MC Tacconi[1], Sophie Badie[1,†], Giuliana De Gregoriis[1,†], Timo Reisländer[1,†] (iD), Xianning Lai[1], Manuela Porru[2], Cecilia Folio[1], John Moore[3], Arnaud Kopp[4], Júlia Baguña Torres[5], Deborah Sneddon[5], Marcus Green[3], Simon Dedic[1], Jonathan W Lee[1], Ankita Sati Batra[6], Oscar M Rueda[6], Alejandra Bruna[6], Carlo Leonetti[2], Carlos Caldas[6], Bart Cornelissen[5], Laurent Brino[4], Anderson Ryan[3], Annamaria Biroccio[2,*] (iD) & Madalena Tarsounas[1,**] (iD)

## Abstract

Due to compromised homologous recombination (HR) repair, *BRCA1*- and *BRCA2*-mutated tumours accumulate DNA damage and genomic rearrangements conducive of tumour progression. To identify drugs that target specifically BRCA2-deficient cells, we screened a chemical library containing compounds in clinical use. The top hit was chlorambucil, a bifunctional alkylating agent used for the treatment of chronic lymphocytic leukaemia (CLL). We establish that chlorambucil is specifically toxic to BRCA1/2-deficient cells, including olaparib-resistant and cisplatin-resistant ones, suggesting the potential clinical use of chlorambucil against disease which has become resistant to these drugs. Additionally, chlorambucil eradicates BRCA2-deficient xenografts and inhibits growth of olaparib-resistant patient-derived tumour xenografts (PDTXs). We demonstrate that chlorambucil inflicts replication-associated DNA double-strand breaks (DSBs), similarly to cisplatin, and we identify ATR, FANCD2 and the SNM1A nuclease as determinants of sensitivity to both drugs. Importantly, chlorambucil is substantially less toxic to normal cells and tissues *in vitro* and *in vivo* relative to cisplatin. Because chlorambucil and cisplatin are equally effective inhibitors of BRCA2-compromised tumours, our results indicate that chlorambucil has a higher therapeutic index than cisplatin in targeting BRCA-deficient tumours.

**Keywords** alkylating agents; BRCA1; BRCA2; cisplatin; DNA damage responses
**Subject Categories** Cancer; Pharmacology & Drug Discovery

## Introduction

*BRCA1* and *BRCA2* germline mutations have been associated with approximately 25% of the familial cases of breast and ovarian cancer (Futreal *et al*, 1994; Miki *et al*, 1994; Wooster *et al*, 1995); therefore, *BRCA1* and *BRCA2* represent classical tumour suppressor genes (Lord & Ashworth, 2016). In addition, somatic *BRCA1* and *BRCA2* mutations, as well as their epigenetic inactivation, have been unravelled in a significant proportion of the sporadic cancers, by recent comprehensive genome sequencing studies (Cancer Genome Atlas Research Network, 2011; Cancer Genome Atlas Network, 2012; Curtis *et al*, 2012; Ali *et al*, 2014; Pereira *et al*, 2016). Thus, the subset of patients affected by *BRCA1/2* mutations appears to be greater than initially anticipated.

BRCA1 and BRCA2 play essential roles in DNA replication and DSB repair (Michl *et al*, 2016). Both factors promote HR, a DNA repair pathway active during S/G2 phases of the cell cycle, which also provides a mechanism for the re-start of stalled replication forks. Consequently, *BRCA1* or *BRCA2* abrogation confers exquisite sensitivity to DNA damage-inducing drugs, in particular those inflicting cytotoxic DNA crosslinks (i.e. platinum drugs and DNA alkylators), which interfere with DNA replication.

Sensitivity of *BRCA1/2*-mutated tumours to platinum compounds has been validated in multiple pre-clinical and clinical studies (Byrski *et al*, 2009, 2010; Silver *et al*, 2010; Tutt *et al*, 2018). Cisplatin and its derivatives are widely used chemotherapeutic drugs, which inflict complex DNA lesions in the form of intra- and inter-strand crosslinks (ICLs; Deans & West, 2011). Similar lesions are induced by DNA-alkylating agents (Fu *et al*, 2012), which include mono-functional (e.g. mitomycin C, nimustine) or bifunctional alkylators (e.g. chlorambucil, cyclophosphamide, melphalan),

1 Genome Stability and Tumorigenesis Group, Department of Oncology, The CR-UK/MRC Oxford Institute for Radiation Oncology, University of Oxford, Oxford, UK
2 Area of Translational Research, IRCCS Regina Elena National Cancer Institute, Rome, Italy
3 Lung Cancer Translational Science Research Group, Department of Oncology, The CR-UK/MRC Oxford Institute for Radiation Oncology, University of Oxford, Oxford, UK
4 Institut de Génétique et de Biologie Cellulaire et Moléculaire (IGBMC), Inserm U1258, CNRS (UMR 7104), Université de Strasbourg, Illkirch, France
5 Radiopharmaceuticals and Molecular Imaging Group, Department of Oncology, The CR-UK/MRC Oxford Institute for Radiation Oncology, University of Oxford, Oxford, UK
6 Department of Oncology, Cancer Research UK Cambridge Institute, University of Cambridge, Cambridge, UK
*Corresponding author. Tel: +39 06 5266 2569 2545; E-mail: annamaria.biroccio@ifo.gov.it
**Corresponding author. Tel: +44 1865 617319; E-mail: madalena.tarsounas@oncology.ox.ac.uk
†These authors contributed equally to this work

some showing specific toxicity against BRCA1/2-deficient cells and tumours (Evers *et al*, 2010; Vollebergh *et al*, 2014; Pajic *et al*, 2017). Interestingly, cisplatin induces primarily intrastrand cross-links (Jamieson & Lippard, 1999), whilst bifunctional alkylators cause mainly ICLs, which represent the most potent type of cyto-toxic DNA lesion (McHugh *et al*, 2001). Although alkylating agents display similar selectivity to cisplatin in targeting BRCA1/2-deficien-cies, they have largely been abandoned for clinical use in breast and ovarian cancers, due to early sub-optimal results in non-stratified patient populations (Williams *et al*, 1985).

Small molecule inhibitors of poly(ADP-ribose) polymerase (PARP) are currently at the forefront of clinical research for the treatment of BRCA-compromised breast, ovarian and prostate tumours (Mateo *et al*, 2015; Mirza *et al*, 2016; Robson *et al*, 2017; Litton *et al*, 2018). PARP inhibitors induce DNA damage indirectly (Lord & Ashworth, 2017) by immobilising PARP enzymes to DNA ends and suppressing their ability to PARylate various substrates (Murai *et al*, 2012; Pascal & Ellenberger, 2015).

In spite of the fact that platinum drugs and PARP inhibitors show initially good responses in the clinic, most patients acquire resis-tance to these drugs (Rottenberg *et al*, 2007; Sakai *et al*, 2008; Shafee *et al*, 2008; Tutt *et al*, 2010; Norquist *et al*, 2011; Ter Brugge *et al*, 2016). Thus, there is a clear necessity for identifying new drugs or drug combinations that can target BRCA1/2-deficient cells and tumours. Here, we report the screen of a chemical library containing 1,280 drugs approved for clinical use by the US Food and Drug Administration (FDA). The highest scoring hit in our screen was chlorambucil, a bifunctional alkylator routinely used in chemotherapeutic regimens against CLL (Goede *et al*, 2014; Jain & O'Brien, 2015). We demonstrate that chlorambucil has high selec-tive toxicity against human cells and xenograft tumours with compromised BRCA1/2 function. Mechanistically, chlorambucil acts by inducing replication stress and DSBs in actively replicating cells. Although similar to cisplatin in targeting BRCA-deficient tumours, chlorambucil shows substantially lower toxicity to normal cells and tissues. Our results suggest that the clinical use of chlorambucil in the BRCA1/2-deficient subset of cancer patients should be re-evaluated.

# Results

## Pharmacological screen for drugs that selectively eliminate BRCA2-deficient cells

In order to identify drugs in clinical use that can target specifically BRCA2-deficient cells, we performed a viability screen using the Prestwick chemical library (http://www.prestwickchemical.com/lib raries-screening-lib-pcl.html) containing 1,280 FDA-approved drugs. Since all drugs are suitable for human testing, any compounds identified in this screen could rapidly be repurposed for the treat-ment of BRCA1/2-mutated patients. We conducted two independent screens, each in triplicate, at drug concentration of 5 µM (Dataset EV1, Appendix Fig S1) using hamster BRCA2-deficient VC8 cells and control BRCA2-complemented cells (Kraak-man-van der Zwet *et al*, 2002). We demonstrated previously (Chai-kuad *et al*, 2014; Zimmer *et al*, 2016) that these BRCA2-deficient cells are hypersensitive to PARP inhibitors, ERK1/2 inhibitors and

pyridostatin, when compared to BRCA2-proficient counterparts. A similar chemical library screen aiming to identify drugs that target BRCA2-deficiency was previously performed (Evers *et al*, 2010) using $Brca2^{-/-}$ mouse mammary tumour-derived cell lines and the LO-PAC®1280 Sigma library of pharmacologically active compounds (Dataset EV1). The chemical composition of this library was dif-ferent from that of the Prestwick library used here, with the two libraries having approximately 25% compounds in common.

Among the top scoring hits in our Prestwick library screens (Dataset EV1, Appendix Fig S1), we identified chlorambucil, a bifunctional alkylating agent used in the past for the treatment of breast and ovarian cancer (Williams *et al*, 1985; Senn *et al*, 1997), irinotecan, a topoisomerase I inhibitor in use for the treatment of cancer patients with *BRCA1* mutations (Kennedy *et al*, 2004), and disulfiram, an aldehyde dehydrogenase inhibitor used in the clinic as an alcohol deterrent. Our group has recently characterised disulfi-ram as an agent specifically toxic to BRCA1/2-deficient cells and tumours, with significant therapeutic potential (Tacconi *et al*, 2017).

Given that our screens were conducted in hamster cells, we vali-dated chlorambucil and irinotecan in BRCA2-deficient human cells. Human colorectal adenocarcinoma $BRCA2^{-/-}$ DLD1 cells (Zimmer *et al*, 2016; Fig 1A) were hypersensitive to both drugs, when compared with $BRCA2^{+/+}$ DLD1 cells. Olaparib and cisplatin were used as controls for selective targeting of BRCA2-deficient cells. Moreover, spheroid cultures established from $BRCA2^{-/-}$ DLD1 cells recapitulated the chlorambucil sensitivity observed in 2D cultures (Fig 1B).

## Chlorambucil is toxic to BRCA1-deficient tumour cells, including those that acquired olaparib resistance

To address the efficacy of chlorambucil against other HR-deficient cells, we assessed the response to this drug in BRCA1-deficient human cells. RPE1 cells immortalised by hTERT overexpression and *TP53* knockout, which carry a *BRCA1* CRISPR/Cas9-mediated dele-tion (Zimmermann *et al*, 2018), were hypersensitive to chlorambu-cil, as well as to olaparib and cisplatin used as controls (Fig 2A).

Moreover, we tested chlorambucil in cellular models in which *BRCA1* gene inactivation is associated with olaparib resistance. Olaparib sensitivity characteristic of *Brca1*-deleted mouse mammary tumour-derived cells is abrogated upon loss of 53BP1 (Fig 2B; Bouwman *et al*, 2010; Tacconi *et al*, 2017). Nevertheless, these cells remained hypersensitive to cisplatin and chlorambucil. Notably, $Brca1^{-/-}53bp1^{-/-}$ cells were more sensitive to cisplatin than $Brca1^{-/-}$ cells. A similar trend was previously reported in mouse embryonic fibroblasts (Bunting *et al*, 2012), suggesting that BRCA1/ 53BP1-deficient, olaparib-resistant tumours may be also more responsive to cisplatin in the clinic. This is indeed the case as demonstrated by a recent clinical trial in which patients with *BRCA1/2* mutated, PARP inhibitor-resistant ovarian cancers showed a robust response to platinum-based therapies (Ang *et al*, 2013).

To generate a second model of olaparib resistance, we inacti-vated REV7 using two different shRNAs in *Brca1*-deleted mouse cells, as previously described (Xu *et al*, 2015). Cells lacking both REV7 and BRCA1 were less sensitive to olaparib than BRCA1-deficient; however, they were effectively eliminated by cisplatin and chlorambucil treatments (Fig 2C). Thus, chlorambucil, similarly to

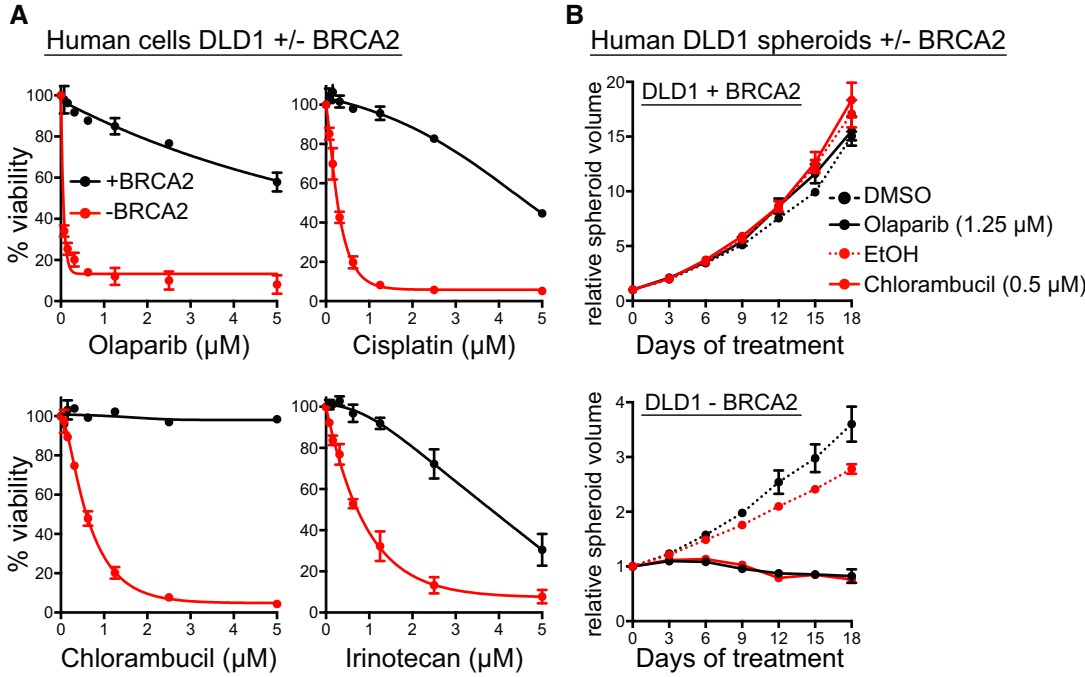

**Figure 1. Chlorambucil sensitivity of BRCA2-deficient human cells and spheroids.**

A   Dose-dependent viability assays of BRCA2-proficient (+BRCA2) or BRCA2-deficient (−BRCA2) human DLD1 cells treated with drugs at the indicated concentrations for 6 days.

B   Human spheroids established from BRCA2-proficient (+BRCA2) or BRCA2-deficient (−BRCA2) DLD1 cells were incubated with 1.25 μM olaparib or 0.5 μM chlorambucil over the indicated period of time.

Data information: (A, B) Graphs represent average values obtained from three independent experiments, each performed in triplicate. Error bars represent SEM.

cisplatin, can eliminate BRCA1-deficient cells that developed PARP inhibitor resistance via 53BP1 or REV7 inactivation.

**Cisplatin-resistant BRCA2-deficient cells derived from human tumours are targeted by chlorambucil**

To further investigate the therapeutic potential of chlorambucil, we tested its effect in cell lines established from BRCA2-compromised human tumours. Capan-1 cells derived from a pancreatic adenocarcinoma carry a C-terminal BRCA2 truncation, which impairs RAD51 nuclear localisation (Chen *et al*, 1998). Capan-1 cells showed significantly higher sensitivity to chlorambucil, as well as to cisplatin and olaparib, when compared to MIA PaCa-2 pancreatic cancer cells with normal BRCA2 expression (Fig 3A).

As a second tumour-derived model, we used PEO1 cells established from a human ovarian tumour carrying a N-terminal BRCA2 truncation, which abrogates HR repair. C4-2 cells, in which wild-type *BRCA2* was restored by treatment with cisplatin (Sakai *et al*, 2009), were used as a control. Viability assays demonstrated that PEO1 cells were hypersensitive to chlorambucil, in contrast to C4-2 cells (Fig 3B). Notably, PEO1 cells, as well as other human cell lines lacking BRCA1 or BRCA2 (DLD1 *BRCA2*$^{-/-}$, HCT116 *BRCA2*$^{-/-}$ and RPE1 *BRCA1*$^{-/-}$), showed sensitivity to melphalan, another bifunctional alkylator (Appendix Fig S2). These results support the efficacy of other bifunctional alkylators against BRCA1/2-deficient cells, in agreement with previous studies (Evers *et al*, 2010).

Loss of the chromatin remodelling factor CHD4 confers resistance to cisplatin in BRCA2-deficient PEO1 cells, through unknown, HR-independent mechanisms that confer DNA damage tolerance (Guillemette *et al*, 2015). We recapitulated this observation by inhibiting CHD4 expression in PEO1 BRCA2-deficient cells (Fig 3C). Lentiviral shRNA-mediated CHD4 depletion increased resistance of PEO1 cells to cisplatin, whilst it had no effect on the cisplatin response of BRCA2-proficient C4-2 cells. Importantly, chlorambucil effectively eliminated both cisplatin-sensitive and cisplatin-resistant BRCA2-deficient PEO1 cells. These results suggest a potential clinical use for chlorambucil in targeting BRCA2-deficient tumours which acquired cisplatin resistance.

**Chlorambucil induces replication stress and DNA damage accumulation in BRCA2-deficient cells**

Alkylating agents can inflict DNA lesions in the form of intra- and inter-strand DNA crosslinks, with a bias towards the latter (Deans & West, 2011). HR repair is an obligatory step in ICL resolution. In cells with compromised HR repair, ICLs interfere with DNA replication, leading to DSB accumulation and cell death (Michl *et al*, 2016). We therefore addressed the possibility that chlorambucil toxicity to BRCA2-deficient cells is due to ICL-inflicted DNA replication and DSB repair defects. The response of BRCA2-proficient and BRCA2-deficient DLD1 cells to chlorambucil was evaluated using time course experiments and immunoblotting

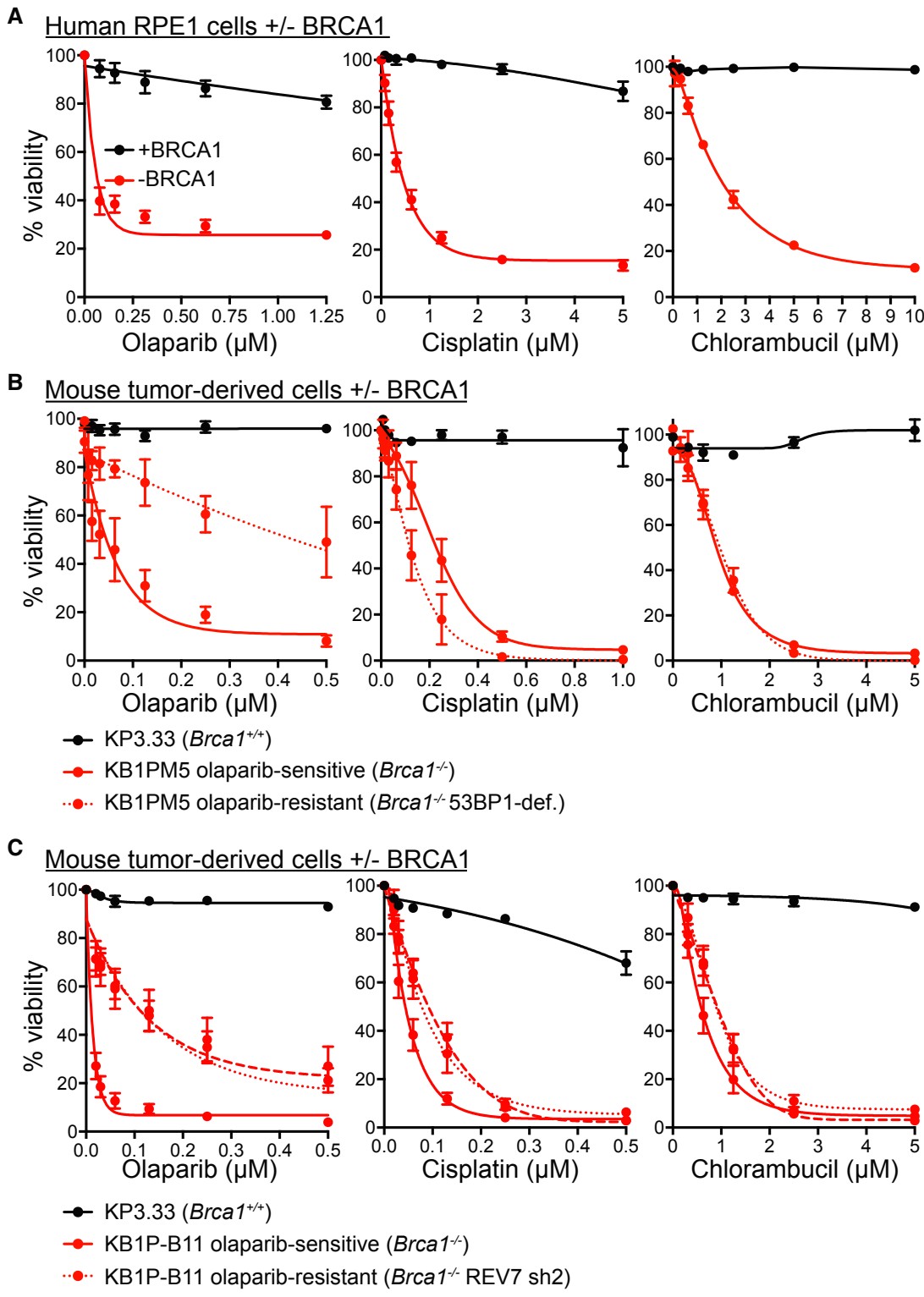

**Figure 2. Chlorambucil sensitivity of BRCA1-deficient human and mouse cells, including those that have acquired olaparib resistance.**

A    Dose-dependent viability assays of BRCA1-proficient (+BRCA1) or BRCA2-deficient (−BRCA1) human RPE1-hTERT and *TP53*-deleted cells treated with drugs at the indicated concentrations for 6 days.

B, C    Dose-dependent viability assays of *Brca1*[+/+] and *Brca1*[−/−] mouse mammary tumour-derived cell lines treated with drugs at the indicated concentrations for 6 days.

Data information: (A–C) Graphs represent average values obtained from three independent experiments, each performed in triplicate. Error bars represent SEM.

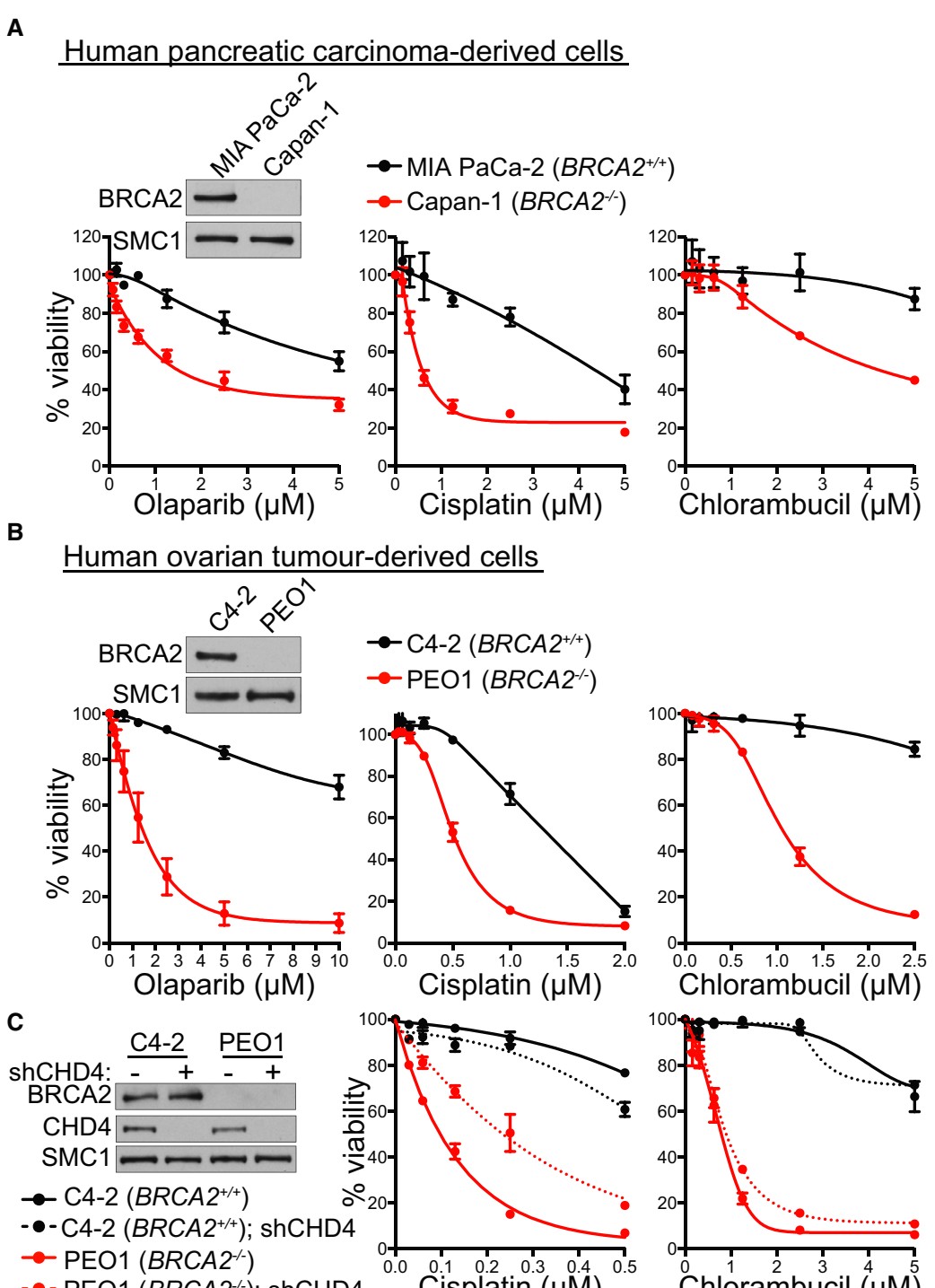

**Figure 3. Chlorambucil sensitivity of BRCA2-deficient human tumour-derived cell lines, including those that have acquired cisplatin resistance.**

A   Dose-dependent viability assays of BRCA2-deficient (Capan-1) or BRCA2-proficient (MIA PaCa-2) human pancreatic carcinoma-derived cells treated with drugs at the indicated concentrations for 6 days.

B   Dose-dependent viability assays of BRCA2-deficient (PEO1) or BRCA2-proficient (C4-2) human ovarian tumour-derived cells treated with drugs at the indicated concentrations for 6 days.

C   BRCA2-deficient (PEO1) or BRCA2-proficient (C4-2) human ovarian tumour-derived cells were infected with lentiviruses expressing control or CHD4 shRNAs, followed by selection with puromycin for 72 h. Dose-dependent viability assays were performed on cells treated with drugs at the indicated concentrations for 6 days.

Data information: (A, B) Graphs represent average values obtained from three independent experiments, each performed in triplicate. Error bars represent SEM. (C) Error bars represent SEM of three technical replicates.

Source data are available online for this figure.

for checkpoint activation markers (Fig 4A). RPA phosphorylation at Ser33, a marker for ATR activation and replication stress (Zeman & Cimprich, 2014), was induced in BRCA2-proficient cells following exposure to 1 μM chlorambucil for 48 h. In contrast, BRCA2-deficient cells, with intrinsic defects in replication fork progression and stability (Zimmer et al, 2016), showed detectable levels of RPA Ser33 phosphorylation even in the absence of any treatment (0 h), and these were markedly increased upon incubation with 1 μM chlorambucil from 16 h onwards. BRCA2-deficient cells also showed elevated levels of KAP1 Ser824 phosphorylation, a signature of ATM-dependent checkpoint activation and DNA damage accumulation. Phosphorylated KAP1 was detected from 24 h of treatment with chlorambucil in BRCA2-deficient cells. As RPA phosphorylation occurs earlier (16 h), this suggests that replication stress may precede DSB formation in response to chlorambucil. Cleaved PARP, an apoptosis marker, was induced only in cells lacking BRCA2. PARP cleavage was detectable from 16 h onwards, similarly to Ser33 RPA phosphorylation, supporting the concept that replication stress underlies chlorambucil toxicity to BRCA2-deficient cells.

ATM activation occurs in response to DSB accumulation and leads to cytotoxicity. Therefore, we next quantified the frequency of DSBs and chromosome aberrations in $BRCA2^{+/+}$ and $BRCA2^{-/-}$ cells upon treatment with 1 μM cisplatin or 1 μM chlorambucil for 72 h (Fig 4B). This concentration was chosen for both drugs because it has the least toxic effects against BRCA2-proficient cells in viability assays (Fig 1A), whilst it induced apoptosis, measured by PARP cleavage (Fig 4A and Appendix Fig S3A), in BRCA2-deficient cells. Both treatments inflicted a significant level of DSBs and chromosome aberrations in BRCA2-deficient cells (Fig 4B), with cisplatin inducing more lesions than chlorambucil. This reflects the higher cytotoxicity of cisplatin (Fig 1A), as only 10% of $BRCA2^{-/-}$ cells remain viable upon exposure to 1 μM cisplatin, in contrast to 30% of the cells treated with 1 μM chlorambucil. More than 90% of $BRCA2^{+/+}$ cells are viable after treatment with 1 μM of either drug. DNA damage accumulation leads to checkpoint activation which alters cell cycle progression. Consistent with this, we observed that a high percentage of cisplatin-treated cells arrested in G2/M (Fig 4C). In contrast, chlorambucil treatment did not cause a significant G2/M arrest, in spite of inducing DNA damage associated with KAP1 phosphorylation (Fig 4A).

## Molecular determinants of sensitivity to chlorambucil and cisplatin

Elucidating which pathways are involved in the repair of chlorambucil- and cisplatin-induced DNA lesions is essential for understanding the mechanism of action of these drugs. This is particularly important for cancer chemotherapy, as resistance is often associated with enhanced DNA repair. On the other hand, defining novel vulnerabilities to chemotherapeutic drugs may provide means to sensitise resistant tumours through novel combination therapies more active in the clinic. We therefore used siRNA depletion in human RPE1 cells to identify DNA damage response factors whose inactivation sensitises cells to chlorambucil and/or cisplatin. Both drugs induce Ser33 RPA phosphorylation, indicative of replication stress and ATR

activation (Fig 4A and Appendix Fig S3A). We thus depleted ATR using siRNA and observed that cells lacking ATR are hypersensitive to both drugs, supporting a key role for this checkpoint kinase in the cellular responses to chlorambucil and cisplatin (Fig 4D). This result is consistent with a recent study, which showed that depletion of CHK1, an ATR phosphorylation target, sensitises cells to chlorambucil and cisplatin (Bruno et al, 2017).

ATR orchestrates cell responses to replication stress, including activation of the Fanconi anaemia (FA) pathway of ICL recognition and repair (Zhang & Walter, 2014). Given the ability of cisplatin and chlorambucil to induce ICLs, it was perhaps not surprising that abrogating FANCD2, a central FA protein, sensitised human cells to both agents (Fig 4E). ATR regulates FA by promoting FANCD2 mono-ubiquitylation (Andreassen et al, 2004), which, in turn, recruits to the chromatin nucleases required for ICL incision and unhooking. The XPF-ERCC1 nuclease makes the first incision at ICL sites. SNM1A nuclease is then recruited to those structures that resemble stalled forks, to complete ICL unhooking in concert with XPF-ERCC1 (Abdullah et al, 2017). Hypersensitivity of ERCC1 and XPF mutant cells to crosslinking anti-cancer drugs is well documented (McHugh et al, 2001); however, whether SNM1A-deficient cells recapitulate this sensitivity is unknown. We found that SMN1A-depleted human cells were also sensitive to cisplatin and chlorambucil (Fig 4E), indicating an important role for this nuclease in ICL repair.

Whilst XPF-ERCC1 is a known downstream effector of the FA pathway, whether SNM1A functions in FA-dependent manner is unknown. To address a possible cooperation between FA and SNM1A in ICL repair, we co-depleted FANCD2 and SNM1A and evaluated the response to cisplatin and chlorambucil. We found that inactivation of both factors did not further sensitise cells compared to FANCD2 depletion alone (Fig 4E). Control experiments performed in cells depleted of XPF and/or FANCD2 using siRNAs showed a similar pattern (Appendix Fig S4). These results demonstrate the concerted action of SNM1A, XPF and FANCD2 upon ICL induction and place for the first time the SNM1A nuclease within the FA pathway of ICL repair.

## The anti-tumoral effect of chlorambucil against BRCA2-deficient xenografts is similar to cisplatin

Chlorambucil inhibited specifically the growth of BRCA2-deficient spheroids established from DLD1 cells (Fig 1B), indicative of its potential use in a tumour setting. Thus, we used $BRCA2^{+/+}$ and $BRCA2^{-/-}$ DLD1 cells to generate xenograft tumours in mice (Fig 5A and B). Chlorambucil had no effect on the growth of BRCA2-proficient tumours (Fig 5A), but it caused a striking reduction in BRCA2-deficient tumour growth (Fig 5B). When the drug was administered intraperitoneally at doses of 3 mg/kg daily for 10 days (with a 2-day break after day 5), we observed tumour eradication in all animals within 21 days from treatment initiation.

Chlorambucil doses approved for CLL patient treatment are 0.2–1.6 mg/kg daily. In our mouse experiments, we used the maximum tolerated dose of 3 mg/kg daily (Weisburger et al, 1975; Grosse et al, 2009), which corresponds to 0.25 mg/kg daily patient dose, calculated using FDA conversion guidelines from mouse to human (http://www.fda.gov/downloads/Drugs/.../Guidances/UCM078932. pdf), and it is therefore clinically relevant. We further reduced the

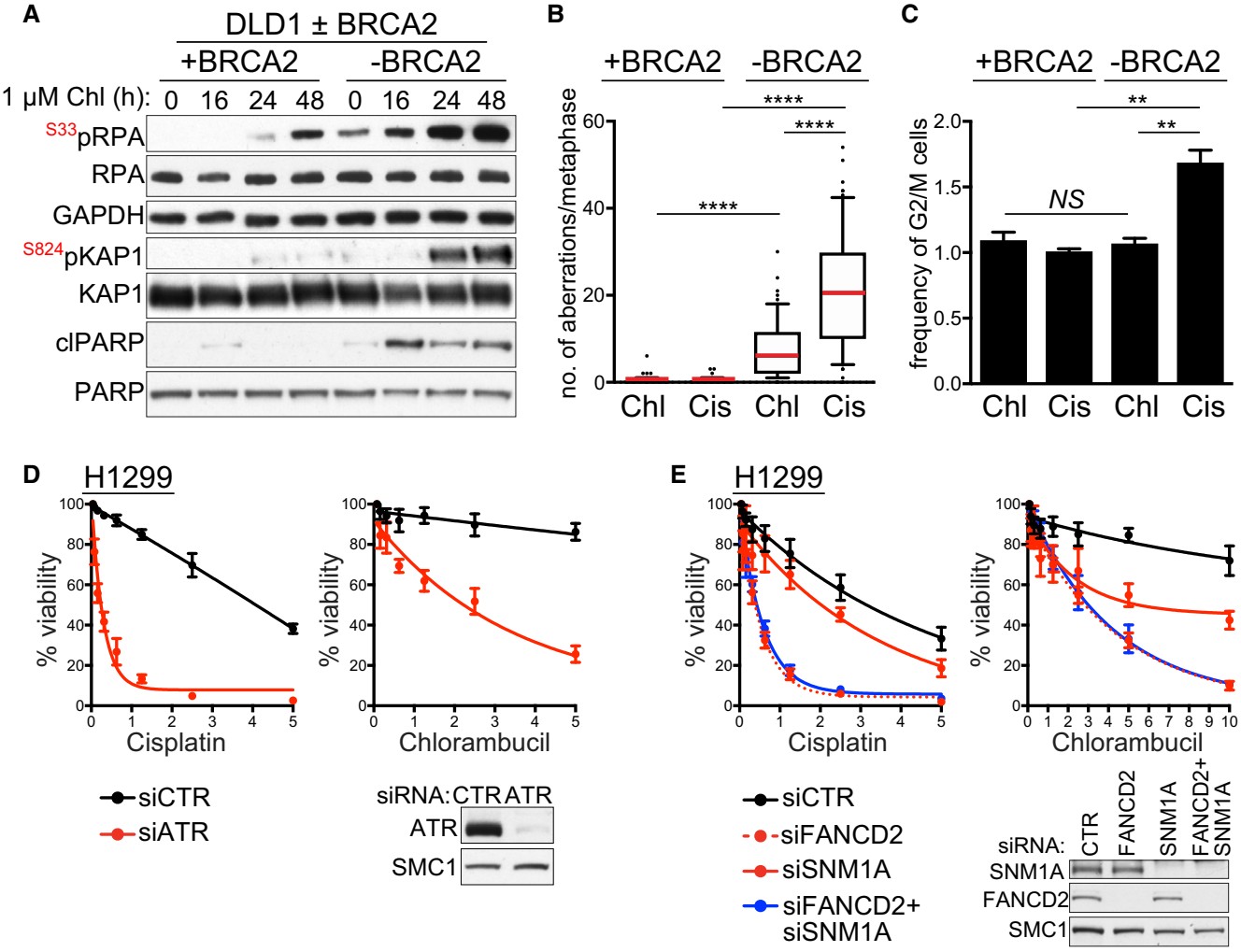

**Figure 4. DNA damage responses to chlorambucil and cisplatin in human cells.**

A  BRCA2-proficient (+BRCA2) or BRCA2-deficient (−BRCA2) human DLD1 cells were incubated with 1 µM chlorambucil (Chl). Whole-cell extracts prepared at the indicated time points during treatment were immunoblotted as shown. GAPDH was used as a loading control.

B  Quantification of chromosome aberrations and chromatid/chromosome break frequencies in BRCA2-proficient (+BRCA2) or BRCA2-deficient (−BRCA2) human DLD1 cells incubated with 1 µM chlorambucil or 1 µM cisplatin for 72 h. Data were obtained from three independent experiments and normalised to untreated controls. A minimum of 60 Giemsa-stained metaphases were analysed for each sample. Cis, cisplatin; Chl, chlorambucil.

C  Quantification of G2/M cell frequency relative to solvent control, using FACS analyses of cells incubated with 1 µM chlorambucil or 1 µM cisplatin for 48 h. Cis, cisplatin; Chl, chlorambucil.

D, E  Human H1299 cells were treated with control (CTR) or indicated siRNAs 2 days before drugs were added to the media for dose-dependent viability assays. Cell extracts prepared at the time of drug addition were immunoblotted as indicated. SMC1 was used as a loading control.

Data information: (B) Whiskers indicate 10–90 percentile, and red bars indicate mean frequencies of breaks. *P*-values were calculated using the Mann–Whitney test. ****$P < 0.0001$. (C) Error bars represent SEM of three independent experiments. *P*-values were calculated using an unpaired two-tailed *t*-test. **$P \leq 0.01$. NS, $P > 0.5$. (D, E) Graphs represent average values obtained from three independent experiments, each performed in triplicate. Error bars represent SEM. Exact *P*-values are included in Appendix Table S1.

Source data are available online for this figure.

dose administered in mice and observed that BRCA2-deficient tumours were also eliminated even at doses of 1 mg/kg daily chlorambucil. However, a 0.3 mg/kg daily dose was ineffective (Appendix Fig S5). Thus, chlorambucil is active against BRCA2-deficient tumours even at doses lower than the equivalent doses used in the clinic.

To further investigate the therapeutic potential of chlorambucil, we used *ex vivo* cultures of patient-derived tumour xenograft cells

(PDTCs; Fig 5C). These recapitulate not only tumour heterogeneity, but also tumour vulnerability to specific drugs (Bruna *et al*, 2016). Chlorambucil was selectively toxic to PDTCs lacking normal BRCA1 expression (STG201, VHIO179; http://caldaslab.cruk.cam.ac.uk/bcape/) and had a small effect on BRCA1-proficient ones (AB521). Importantly, VHIO179, a tumour carrying *BRCA1* germline truncation, is resistant to treatment with PARP inhibitors due to a *MAD2L2 (REV7)* inactivating mutation (Bruna *et al*, 2016; Cruz *et al*, 2018).

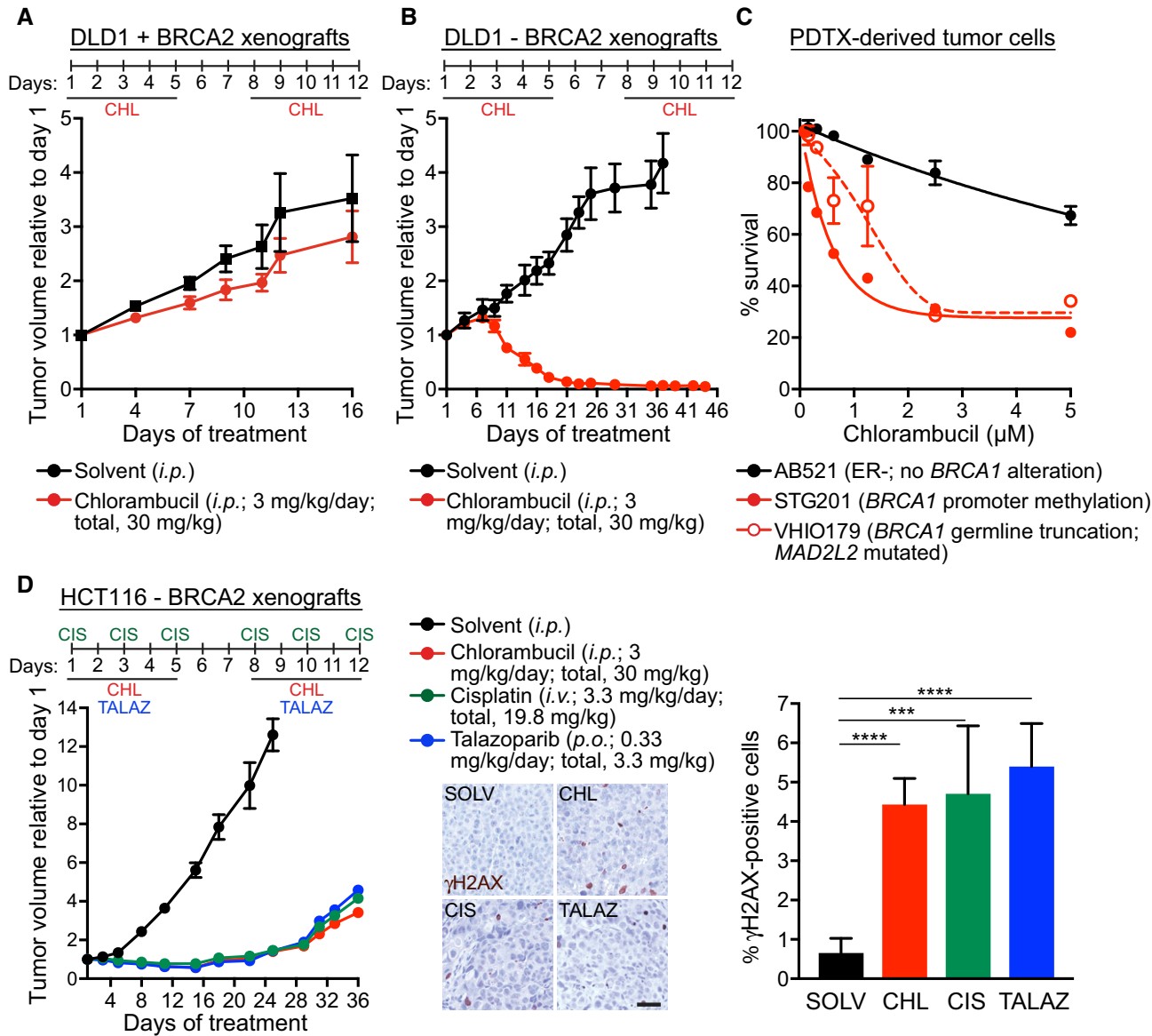

**Figure 5. Chlorambucil impairs growth of BRCA2-deficient tumours and PDTXs.**

A, B   Nude mice (nu/nu) were injected subcutaneously with 5 × 10⁶ human DLD1 cells, BRCA2-proficient (A) or BRCA2-deficient (B). Tumour-bearing mice were treated with 3 mg/kg daily chlorambucil administered intraperitoneally (i.p.) for a total of 10 days. Tumour weight was determined on the indicated days after initiation of the treatment.

C   PDTCs derived from breast cancer samples as previously described (Bruna et al, 2016) were treated with chlorambucil at the indicated doses. Cell survival is represented relative to DMSO control. AB521, ER-negative tumour, no BRCA1 alteration; STG201, tumour with BRCA1 promoter methylation and loss of BRCA1 expression; VHIO179, tumour with BRCA1 germline mutation and MAD2L2 inactivating mutation (olaparib-resistant); http://caldaslab.cruk.cam.ac.uk/bcape/.

D   CB17/SCID mice were injected intramuscularly with 5 × 10⁶ human BRCA2-deficient HCT116 cells. Tumour-bearing mice were treated on the indicated days with chlorambucil or cisplatin administered intraperitoneally (i.p.), or talazoparib administered orally (o.s.) Tumour volume was measured on the indicated days after treatment initiation and was expressed relative to tumour volume at the beginning of treatment (day 1). Scale bar, 40 μm.

Data information: (A, B) Each experimental group included n = 4 mice. Error bars represent SEM. (C) Error bars represent SEM of two independent experiments performed in technical triplicates. (D) Each experimental group included n = 5 mice. Error bars represent SEM. Tumour sections were assessed at the end of each treatment using immunohistochemistry of γH2AX staining. P-values were calculated using an unpaired two-tailed t-test. ****P < 0.0001; ***P < 0.001. Exact P-values are included in Appendix Table S1.

PDTCs derived from this tumour were sensitive to chlorambucil, which supports our results obtained with the *Brca1*-deleted mouse mammary tumour-derived cells, upon REV7 depletion using shRNAs (Fig 2C). These results strengthened the therapeutic potential of

chlorambucil for targeting BRCA-deficient human tumours that acquired resistance to olaparib.

We next determined chlorambucil anti-tumour activity using paired *BRCA2⁺/⁺* and *BRCA2⁻/⁻* HCT116 colon carcinoma cell

lines, generated in one of our laboratories (Xu *et al*, 2014). *BRCA2*$^{-/-}$ HCT116 cells showed hypersensitivity to cisplatin and chlorambucil *in vitro* (Appendix Fig S6A). Xenograft tumours were subsequently established from *BRCA2*$^{+/+}$ and *BRCA2*$^{-/-}$ HCT116 cells and assessed for their response to chlorambucil, cisplatin and talazoparib (Appendix Fig S6B). Chlorambucil had no effect on the BRCA2-proficient tumours, but it inhibited the growth of BRCA2-deficient ones. We further used *BRCA2*$^{-/-}$ HCT116-derived tumours to compare the anti-tumoral effects of cisplatin, chlorambucil and the PARP inhibitor talazoparib (Fig 5D). Treatment with each of the three drugs showed tumour growth inhibition at the nadir of the effect of 89, 89 and 85%, respectively (Table 1), with a progression-free survival of 19, 21 and 24 days, suggesting similar anti-tumoral activities against BRCA2-defective tumours. Importantly, the three drugs inflicted comparable levels of DNA damage evaluated by immunohistochemical staining with an anti-γH2AX antibody (Fig 5D).

### Higher cisplatin toxicity *in vitro* and *in vivo* compared to chlorambucil

Our cell viability assays (Fig 1A) indicated that cisplatin is relatively more toxic to *BRCA2*$^{+/+}$ DLD1 cells than chlorambucil, whilst both drugs induced cleaved PARP expression in *BRCA2*$^{-/-}$ DLD1 cells (Fig 4A and Appendix Fig S3A). To further explore the differential toxicity of the two drugs to BRCA2-proficient cells, we treated MRC5 primary-like cells with cisplatin or chlorambucil for up to 72 h (Appendix Fig S3B). Under these conditions, we observed a higher accumulation of the apoptotic marker in response to cisplatin than chlorambucil treatment, supporting the notion that cisplatin is more toxic than chlorambucil to non-tumour cells.

MRC5 cells have intact p53 pathway (Carlos *et al*, 2013), which may account for the exquisite toxicity of cisplatin and chlorambucil to these cells. p53 is a key determinant of cell sensitivity to platinum-based drugs (Kelland, 2007). We did not succeed in abrogating p53 expression in MRC5 cells, and instead, we used RPE1 cells in which *TP53* was deleted using the CRISPR/Cas9 system to address the role of p53 in the cellular response to chlorambucil. Functional p53 sensitised cells to cisplatin, as well as to chlorambucil (Appendix Fig S3C), whilst its abrogation promoted resistance. This supports the notion that p53-dependent responses mediate the cytotoxicity of these drugs.

The observation that cisplatin is more toxic than chlorambucil *in vitro* prompted us to assess the relative toxicities of the two drugs *in vivo*. We therefore treated wild-type mice with the maximum tolerated doses of each drug and evaluated apoptosis *in vivo* using single-photon emission computed tomography (SPECT) imaging of the apoptosis imaging marker $^{99m}$Tc-Duramycin (Palmieri *et al*, 2018; Fig 6A). SPECT image quantification demonstrated significant accumulation of $^{99m}$Tc-Duramycin in the heart, blood and lungs of cisplatin-treated mice. Apoptosis levels in the organs of chlorambucil-treated mice were similar to the control, solvent-treated group. We furthermore used lung and heart from treated mice for immunohistochemistry staining with γH2AX antibody (Fig 6B). Consistent with the biodistribution of the apoptosis tracer $^{99m}$Tc-Duramycin, we observed significantly higher γH2AX levels in the heart and lungs of cisplatin-treated mice, relative to the chlorambucil-treated ones. DNA lesion accumulation, visualised with γH2AX staining, was also detected in the organs from chlorambucil-treated animals, indicating that both drugs inflict DNA damage *in vivo*. However, cisplatin induced more pronounced and more deleterious lesions than chlorambucil. Given the higher toxicity of cisplatin to normal tissues (Fig 6A and B) and the similar anti-tumour effect of the two drugs (Fig 5C), we propose that chlorambucil represents a potential alternative to cisplatin for the treatment of BRCA-deficient tumours.

## Discussion

In this study, we report identification of the bifunctional alkylator chlorambucil in a chemical library screen for drugs with specific toxicity against BRCA2-deficient cells. DNA alkylators, including chlorambucil, melphalan and nimustine, were also isolated in a previous screen and shown to be active *in vivo* against allografted BRCA2-deleted mouse tumours (Evers *et al*, 2010). Subsequent studies have substantiated the potential of nimustine in targeting BRCA-deficient tumours (Pajic *et al*, 2017).

Our work demonstrates the specific toxicity of chlorambucil to BRCA1/2-deficient human cells and xenograft tumours. Importantly, BRCA1/2-deficient cells with acquired resistance to olaparib or cisplatin show sensitivity to chlorambucil, suggesting its therapeutic potential against this difficult to treat tumour subset. Together with chlorambucil being non-toxic to normal cells and tissues (see

**Table 1.** *In vivo* anti-tumour efficacy of chlorambucil, PARP inhibitor talazoparib and cisplatin on HCT116 BRCA2-deficient xenografts.

| Treatment | Tumour volume inhibition (average, %) | Tumour regression (% mice) | Tumour relapse (% mice) | Median progression-free survival (days; range) | Body weight loss (average, %) | Deaths (% mice) |
|---|---|---|---|---|---|---|
| Chlorambucil | 89 | 100 | 100 | 21 (19–25) | 1 | 0 |
| Talazoparib | 89 | 100 | 100 | 24 (20–24) | 3 | 0 |
| Cisplatin | 85 | 100 | 100 | 19 (18–20) | 7 | 0 |

Tumours were allowed to grow to 250 mm$^3$ before initiation of treatment. Mice were treated with chlorambucil (*i.p.*; 3 mg/kg/day) or talazoparib (*p.o.*; 0.33 mg/kg/day) for five consecutive days, followed by 2-day break and 5 more days of treatment. Cisplatin (*i.p.*; 3.3 mg/kg/day) was administered for three consecutive days, followed by 4-day break and 3 more days of treatment. Each experimental group included *n* = 5 mice. Tumour volume inhibition was calculated at the nadir of the effect using the formula: (1-tumour volume in treated mice/tumour volume in untreated mice) × 100 and expressed as average for *n* = 5 mice in each group. Tumour regression was defined as percentage of mice in which reduction of tumour volume, after the initiation of treatment, was maintained for at least 2 weeks. Tumour relapse was defined as percentage of mice in which tumour regrowth was observed after tumour regression. Median progression-free survival was defined as duration (days) of tumour regression. Body weight loss is reported as weight at the end of treatment relative to the first day of treatment (%), as average for *n* = 5 mice in each group.

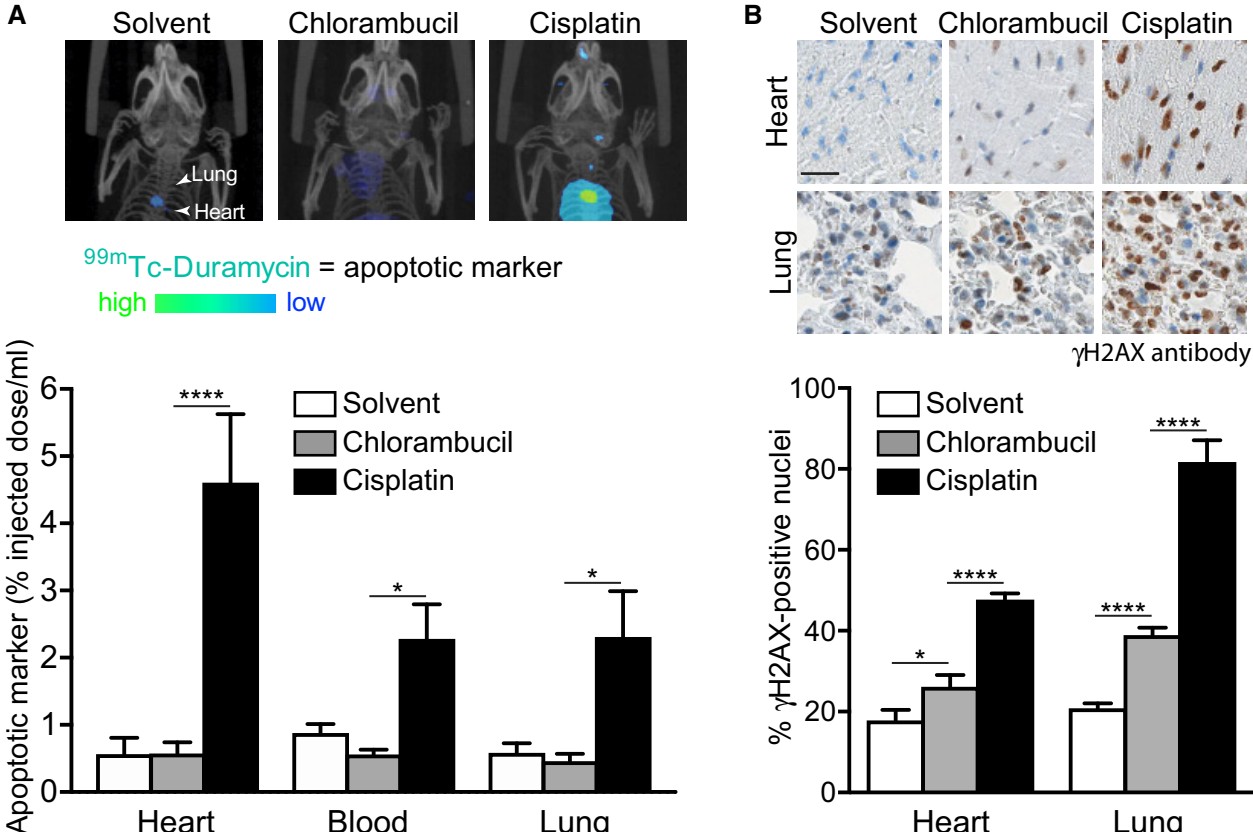

**Figure 6. Chlorambucil vs. cisplatin *in vitro* and *in vivo* toxicity.**

A  Wild-type *Balb/c* mice were injected intraperitoneally with solvent (daily) or 3 mg/kg chlorambucil (daily for 5 days) or 3.3 mg/kg cisplatin (daily for 3 days). Uptake of the apoptosis tracer $^{99m}$Tc-Duramycin 2 h after intravenous injection was quantified in selected organs using SPECT imaging in the indicated organs. Representative maximum intensity partial projections showing tracer distribution are shown.

B  Immunohistochemical analyses of γH2AX staining in organs from mice treated as in (A). Scale bar, 25 μm.

Data information: (A) Each experimental group included $n = 5$ mice. Error bars represent SEM. *P*-values were calculated using one-way ANOVA followed by Tukey's multiple comparisons test. ****$P < 0.0001$; *$P < 0.05$. (B) Organs from $n = 3$ mice were analysed for each treatment. Error bars represent SD. *P*-values were calculated using one-way ANOVA followed by Tukey's multiple comparisons test. ****$P < 0.0001$; *$P < 0.05$. Exact *P*-values are included in Appendix Table S1.

below), these results support chlorambucil re-evaluation in cancer patients with *BRCA* mutations.

Chlorambucil was used in the treatment of breast and ovarian cancer until the late 1970s (Barker & Wiltshaw, 1981; Williams *et al*, 1985; Senn *et al*, 1997). These early studies did not show a significant benefit of chlorambucil, possibly because its effect was obscured by the lack of molecular markers (e.g. *BRCA1/2* status). In some clinical trials, addition of cisplatin to chlorambucil treatment was beneficial (Barker & Wiltshaw, 1981). Although the response to regimens that included cisplatin was initially superior to chlorambucil alone, the overall patient survival was not improved (Williams *et al*, 1985). Nevertheless, chlorambucil-based therapies for breast and ovarian cancer were abandoned, as cisplatin was approved for broader clinical use and became the leading anti-cancer drug and first-line treatment for various malignancies.

A major problem associated with cisplatin chemotherapy is the emergence of drug resistance (Norquist *et al*, 2011). Proposed resistance mechanisms include insufficient drug access to DNA, enhanced DNA repair and apoptotic pathways failure (Siddik, 2003).

Chlorambucil resistance in CLL patients has also been documented (Panasci *et al*, 2001; Norgaard *et al*, 2004), although not to a similar extent as cisplatin resistance. For both drugs, clinical evidence for a unique resistance mechanism is lacking due to the multifaceted response to these drugs in patients. Clearly, both drugs induce ICLs and DSBs; however, the mechanism linking it to apoptosis has not been elucidated. *TP53* mutations known to increase cell tolerance to DNA damage and limit apoptosis can be reliably correlated with cisplatin resistance (Siddik, 2003). We show here that this is also the case for chlorambucil resistance, as indicated by our viability assays using isogenic p53 wild-type and p53-deleted RPE-1 cells.

An important result reported here is that chlorambucil is toxic to cisplatin-resistant BRCA2-deficient cancer cells. Although the mechanism remains to be further elucidated, this may be explained by the fact that the two drugs inflict distinct DNA lesions (chlorambucil induces primarily inter- and cisplatin primarily intrastrand crosslinks) which activate distinct DNA damage response pathways, especially in BRCA-deficient cells with compromised HR repair.

Since cases of cisplatin and chlorambucil resistance (Panasci *et al*, 2001; Norgaard *et al*, 2004; Norquist *et al*, 2011) in tumours are documented, it is important to identify which DNA damage repair defects underlie sensitivity to these drugs. This could lead to novel druggable targets that can sensitise tumours to these agents, with the caveat that combination therapies may increase toxicity. We demonstrate that ATR is a major determinant of sensitivity to chlorambucil and cisplatin. Most likely, DNA crosslinks generated by treatment with the two drugs obstruct DNA replication, and the resulting stalled forks require ATR for their re-start and stabilisation (Zeman & Cimprich, 2014). Furthermore, consistent with the well-established role of FA in ICL repair, we demonstrate that FANCD2 inactivation also sensitises cells to either chlorambucil or cisplatin. As a novel target, we identify SNM1A, a nuclease with a poorly understood function in ICL repair, as a factor required for the proliferation of cells treated with chlorambucil or cisplatin. Chemical inhibitors against this nuclease may re-sensitise resistant tumours to chlorambucil and/or cisplatin. Moreover, our results indicate that SNM1A functions in the context of the FA pathway of ICL repair, which helps elucidate the cellular roles of this nuclease.

A second caveat associated with the clinical use of cisplatin is its well-documented toxicity (Kelland, 2007), mainly in the form of nephropathies and gastrointestinal tract disorders. The problem was partially alleviated by the development of new platinum drugs with lower toxicity, but no benefit for patient survival over cisplatin. Chlorambucil, which has been part of the standard therapy for CLL patients for over 50 years either as single agent or in combination with engineered antibodies (Goede *et al*, 2013, 2014), shows overall mild toxicity, occasionally in the form of pancytopenia (Rai *et al*, 2000).

Consistent with the notion that chlorambucil is less toxic than cisplatin in patients, our study demonstrates that *in vitro* and *in vivo* chlorambucil treatment triggers lower levels of apoptosis relative to cisplatin in healthy cells and tissues. We visualised apoptosis in mice using SPECT *in vivo* imaging of a radiolabelled apoptosis tracer and found that it correlates with γH2AX accumulation. IHC staining showed higher γH2AX levels in the organs of cisplatin-treated mice, consistent with cisplatin inflicting more DNA lesions than chlorambucil. To what extent cisplatin toxicity to healthy tissues can be attributed to its ability to induce irreversible intrastrand crosslinks remains to be determined. Importantly, the two drugs were equally effective in suppressing growth of BRCA2-deficient xenografts. Given the lower toxicity of chlorambucil relative to cisplatin in mice and given that the two drugs show comparable anti-tumoral activity against BRCA-deficient tumours, our results suggest that chlorambucil is a drug with a therapeutic index superior to cisplatin against the BRCA-deficient tumour subset. In addition, chlorambucil is a drug administered orally, like the PARP inhibitor olaparib, in contrast to cisplatin which is an intravenous drug.

One potential caveat associated with administering alkylating agents, including chlorambucil, to CLL patients is the relatively high risk of primary tumours at other sites (Maurer *et al*, 2016). Whilst concrete evidence for this association was lacking in an initial study (Hisada *et al*, 2001), a subsequent evaluation established an increased (but not significant) incidence of epithelial cancers and acute myeloid leukaemia in chlorambucil recipients (Grosse *et al*, 2009). Thus, potential carcinogenic effects of chlorambucil must be taken into consideration when evaluating chlorambucil and other agents known to inflict ICLs for clinical use.

Overall the results reported here suggest that the efficacy of chlorambucil assessed in early clinical trials of breast and ovarian cancer patients was obscured by the lack of molecular markers. Thus, patient stratification based on *BRCA1/2* status may help identify those tumours vulnerable to chlorambucil. Recent studies have shown that treatments based on the alkylating agent cyclophosphamide are effective in the subset of BRCA1/2-mutated patients (Vollebergh *et al*, 2014). Our results corroborate the specificity of alkylating agents in this setting and suggest that the therapeutic potential of chlorambucil in *BRCA1/2*-mutated patients should be re-evaluated.

## Materials and Methods

### Cell lines and growth conditions

*BRCA2*-mutated hamster cells transduced with empty vector or BRCA2 [V-C8 and V-C8 + BRCA2, respectively (Kraakman-van der Zwet *et al*, 2002)], $BRCA2^{+/+}$ and $BRCA2^{-/-}$ human colorectal adenocarcinoma DLD1 cells (Horizon Discovery; Zimmer *et al*, 2016), MIA PaCa-2 human pancreatic carcinoma cells and MRC5 human lung fibroblast cells were cultivated in monolayers in DMEM (Sigma-Aldrich) supplemented with 10% foetal bovine serum (Life Technologies) and 1% penicillin/streptomycin (Sigma-Aldrich). Human retinal pigment epithelial cells RPE1, wild type or transduced with hTERT and *TP53*-deleted ($BRCA1^{+/+}$ and $BRCA1^{-/-}$; a gift from Dr. Dan Durocher, University of Toronto, Canada; Zimmermann *et al*, 2018) were cultivated as above in presence of 2 μg/ml blasticidin (Life Technologies). $BRCA2^{+/+}$ and $BRCA2^{-/-}$ human colorectal carcinoma HCT116 cells (Ximbio, Cancer Research Technology) were grown in McCoy's 5a media (Life Technologies) with 10% foetal bovine serum and 1% penicillin/streptomycin.

Mouse mammary tumour cell lines KP3.33 ($Brca1^{+/+}$ control), KB1PM5 ($Brca1^{-/-}$, PARP inhibitor sensitive) and KB1PM5 [$Brca1^{-/-}$, 53BP1-deficient, PARP inhibitor resistant (Jaspers *et al*, 2013)] were cultured at 37°C, 5% $CO_2$ and 3% oxygen in complete medium DMEM/F-12 (Life Technologies) supplemented with 10% foetal bovine serum (Life Technologies), 1% penicillin/streptomycin (Sigma-Aldrich), 5 μg/ml insulin (Sigma-Aldrich), 5 ng/ml epidermal growth factor (Life Technologies) and 5 ng/ml cholera toxin (Gentaur).

Human Capan-1 pancreatic carcinoma-derived cells were cultivated in IMDM (Life Technologies) with 20% foetal bovine serum, 1% penicillin/streptomycin. Human PEO-1 ovarian cancer cells and the BRCA2-restored clone C4-2 (Sakai *et al*, 2009) were grown in RPMI (Life Technologies) supplemented with 2 mM sodium pyruvate and 10% foetal bovine serum (Life Technologies). All cell lines used in this study are p53-compromised, with the exception of human MRC5 and human RPE1 wild-type cells used for Appendix Fig S3B and C. All cell lines were routinely genotyped and tested for mycoplasma contamination.

Chlorambucil (Abcam), irinotecan hydrochloride (Cambridge Bioscience Ltd), melphalan (Bio-Techne R&D Systems), cisplatin (Sigma-Aldrich) and olaparib (Selleckchem) were added to the media at the concentrations indicated. Cells were arrested in

mitosis with 0.2 µg/ml KaryoMAX® Colcemid (Life Technologies) for 16 h.

### Prestwick chemical library screen setup and statistical analysis

Two independent screens were performed, each in triplicate. BRCA2-deficient and BRCA2-reconstituted hamster cells, respectively, V-C8 and V-C8 + BRCA2 (Kraakman-van der Zwet et al, 2002), were seeded in 96-well plates. Drugs of the Prestwick Chemical Library (http://www.prestwickchemical.com/prestwick-chemical-library.html) supplied at 10 mM in DMSO were added 24 h later at 5 µM dilution in 100 µl of culture media. Each plate contained DMSO control wells. Following incubation with the drugs for 3 days (screen 1) or 6 days (screen 2), viability was assessed using resazurin-based assays.

The library consisted of 16 plates (32 plates were used per cell line in each screen); therefore, a median plate normalisation procedure was applied, corresponding to a modified version of the robust per cent of sample (Birmingham et al, 2009). Hits were ranked using strictly standardised mean deviation (Dataset EV1; Zhang, 2011).

### Cell viability assays

Cells were plated at densities varying between 100 and 2,000 cells per well in 96-well plates. These densities were determined for each cell line individually, so that they reached 80–90% confluency after 7 days in culture in the absence of any treatment. Drugs were added at the indicated concentrations on the following day. Six days later, cell viability was determined by incubating cells with medium containing 10 µg/ml of resazurin for 2 h. Fluorescence was measured at 590 nm using a plate reader (POLARstar, Omega). Cell viability was expressed relative to cells treated with vehicle control of the same cell line, thus accounting for any differences in viability caused by genetic modifications.

### Spheroid cultures

Ten thousand cells of colorectal adenocarcinoma DLD1 cells were plated in a well of a 96 round-bottom well, ultra-low attachment plate (Costar) before being centrifuged at $210 \times g$ for 10 min. Spheroids were cultured for 4 days as in published protocols (Friedrich et al, 2009) before removal of half of the media and addition of the drugs diluted in that same volume. For untreated control spheroids, complete media containing the corresponding solvent was added. For drug-treated spheroids, fresh media containing the drug were added every 72 h. Pictures of the spheroids were acquired using a Nikon TE2000-E microscope, and volumes were analysed with MATLAB R2014a.

### Immunoblotting

To prepare whole-cell extracts, cells were washed once in 1× PBS, harvested by trypsinisation, washed in 1× PBS and re-suspended in SDS–PAGE loading buffer, supplemented with 0.1 mM DTT. Samples were sonicated using a probe sonicator and heated at 70°C for 10 min. Equal amounts of protein (30–100 µg) were analysed by gel electrophoresis followed by Western blotting. NuPAGE-Novex 10% Bis–Tris and NuPAGE-Novex 3–8% Tris–acetate gels (Life Technologies) were run according to manufacturer's instructions.

### siRNA

RPE1 cells were transfected using DharmaFECT-1 (Dharmacon, #T-2001-03). Briefly, $4 \times 10^5$ cells were transfected with 40 nM siRNA per plate by reverse transfection in 6-cm plates. After 24-h incubation, depletion was determined by immunoblotting. The ATR siRNA sequence was CAG GCA CTA ATT GTT CTT CAA. siRNA SMART pools were used against FANCD2 (Dharmacon, #M-016376-02-0005), XPF/ERCC4 (Dharmacon, #L-019946-00-0005) and against SNM1A (Dharmacon, #M-010790-00-0005). AllStars siRNA (Qiagen, #1027281) was used as a negative control.

### Preparation of metaphase chromosome spreads

Cells synchronised in mitosis via overnight incubation with 0.1 µg/ml KaryoMAX® Colcemid (Life Technologies) were collected by mitotic shake-off and swollen in hypotonic buffer (0.03 M sodium citrate) at 37°C for 25 min. Cells were fixed in freshly prepared 3:1 mix of methanol: glacial acetic acid, and nuclear preparations were dropped onto slides pre-soaked in 45% acetic acid prior to being allowed to dry overnight. The following day, mitotic chromosomes were stained using Giemsa (VWR) and viewed with a Leica DMI6000B inverted microscope equipped with a HCX PL APO 100×/1.4–0.7 oil objective.

### FACS analysis

$BRCA2^{+/+}$ and $BRCA2^{-/-}$ DLD1 cells treated for 48 h with 1 µM of cisplatin or 1 µM chlorambucil were incubated with 10 µM EdU for 1 h at 37°C. Cells were harvested by trypsinisation and processed for EdU staining using the Click-iT Plus Alexa Fluor 647 Flow Cytometry Assay Kit (Thermo Fisher Scientific). Cells were incubated with 20 µg/ml propidium iodide and 10 µg/ml RNase A (Sigma) in PBS. At least 10,000 cells were analysed by flow cytometry (FACSCalibur, BD Biosciences). Data were processed using FlowJo (FlowJo, LLC) software.

### In vivo xenograft experiments

CB17/SCID male mice (CB17/Icr-$Prkdc^{scid}$/IcrIcoCrl, 6 weeks old and weighing 26–28 g) were purchased from Charles River Laboratories (Calco, Italy). The mice were maintained in high-efficiency, particulate air HEPA-filtered racks and were fed autoclaved laboratory rodent diet. All animal procedures were in compliance with the national and international directives (D.L. 4 March 2014, no. 26; directive 2010/63/EU of the European Parliament and of the council; Guide for the Care and Use of Laboratory Animals, United States National Research Council, 2011).

To generate xenografts derived from HCT116 BRCA2-proficient or HCT116 BRCA2-deficient cells, CB17/SCID mice were injected intramuscularly into the hind leg muscles with $5 \times 10^6$ cells per mouse. When a tumour volume of approximately 250 mm³ was evident in BRCA2-proficient (4 days after cell injection) and BRCA2-deficient (6 days after cell injection) xenografts, mice were randomised in vehicle and treated groups and the treatment was initiated. For

xenografts derived from DLD1 BRCA2-proficient or DLD1 BRCA2-deficient cells, tumours were grown subcutaneously until they reached a mean volume of approximately 100 mm$^3$ (BRCA2-deficient) and 150 mm$^3$ (BRCA2-proficient), at which point the treatment was initiated. Each experimental group included five mice.

Chlorambucil (3, 1 or 0.3 mg/kg/day) was administered intraperitoneally for five consecutive days, followed by 2-day break and 5 more days of treatment. Based on previously established maximum tolerated dose (Leonetti *et al*, 1996), cisplatin (3.3 mg/kg/day) was administered intraperitoneally for three consecutive days, followed by 4-day break and 3 more days of treatment. Talazoparib (0.33 mg/kg/day) was administered orally for five consecutive days, followed by 2-day break and 5 more days of treatment. At the time points indicated, tumour volumes were measured in two dimensions using a calliper and tumour weight was estimated from tumour volume (1 mg = 1 mm$^3$). Tumour weight inhibition was calculated using the formula $a \times b^2/2$, where $a$ and $b$ are the long and short sizes of the tumour, respectively, at the nadir of the effect. The number of mice used in each experiment is described in each figure legend.

### *Ex vivo* drug experiments

The *ex vivo* drug treatment protocol was performed as previously described (Bruna *et al*, 2016). Briefly, frozen patient-derived tumour xenografts (PDTXs) were thawed and dissociated on the GentleMACS Dissociator (Miltenyi Biotec, Cat ID 130-093-235) using the Tumour Dissociation Kit, human (Miltenyi Biotec, Cat ID 130-095-929) and preset protocol h_tumour_01. Single cells were plated at ~ 40,000 cells/ml in 96-well plates and dosed 72 h after plating. Cell Titer Glo 3D was added to the cells 6 days after dosing. Plates were read on the Pherastar plate reader using the Luminescence module.

### *In vivo* apoptosis detection

Wild-type Balb/c female mice (6 weeks old and weighing 16–23 g) were purchased from Charles River (UK). Animals were housed in individually ventilated cages in sex-matched groups of up to six per cage in an artificial day–night cycle facility with *ad libitum* access to food and water. All analyses were performed blinded to experimental group assignment.

Mice were injected intraperitoneally with 0.9% saline ($n = 5$), 3.3 mg/kg/day cisplatin ($n = 5$) for 3 days or with 3 mg/kg/day chlorambucil ($n = 5$) for 5 days. $^{99m}$Tc-Duramycin was prepared as previously described (Palmieri *et al*, 2018). The purity was checked by RP18 HPLC and confirmed to be over 95%. Mice were injected intravenously 2 days after the end of treatment with 1 µg of $^{99m}$Tc-Duramycin (2–4 MBq) and were imaged 2 h later, using a VECTor® PET/SPECT/CT scanner (MILabs, Utrecht, the Netherlands). After the final imaging session, mice were euthanised by cervical dislocation and selected organs and tissues were removed. The amount of radioactivity in each organ was measured using a 1470 WIZARD gamma counter (PerkinElmer). Counts per minute were converted into MBq using a calibration curve generated from known standards. These values were decay-corrected to the time of injection, and the percentage of the injected dose per gram (% ID/g) of each organ was calculated. A second cohort of mice ($n = 3$ per group) treated with the same protocol were sacrificed for

### The paper explained

#### Problem

*BRCA1* and *BRCA2* are major tumour suppressors. Germline mutations in these genes are associated with approximately 25% of the familial cases of breast and ovarian cancer and a significant proportion of sporadic cancers show *BRCA1* and *BRCA2* gene inactivation. Thus, there is a large number of patients affected by *BRCA1/2* deficiency. BRCA1 and BRCA2 proteins play key roles in homologous recombination repair and their loss triggers high sensitivity to DNA damage. Drugs that induce DNA lesions are routinely used in the clinic to treat *BRCA1/2*-deficient tumours (e.g. cisplatin and PARP inhibitors). However, most tumours acquire resistance to these therapies and novel strategies for their elimination are needed.

#### Results

Here, we report a screen of the Prestwick chemical library of FDA-approved drugs for compounds that target specifically BRCA2-deficient cells. We identify chlorambucil as the highest scoring hit from this screen, which selectively eliminates BRCA1/2-deficient cells and tumours, including olaparib-resistant and cisplatin-resistant ones. Importantly, chlorambucil is substantially less toxic to normal cells and tissues than cisplatin, a drug routinely used in the clinic for cancer treatment.

#### Impact

Chlorambucil and cisplatin are equally effective in targeting specifically BRCA1/2-deficient tumours. However, chlorambucil exhibits lower toxicity to normal cells. We propose that chlorambucil may provide an effective alternative to cisplatin for the treatment of this tumour subset, either as single therapy or in combination with other agents.

immunohistochemical staining of selected organs. These animal procedures were performed in accordance with the UK Animals (Scientific Procedures) Act 1986 and with local ethical committee approval.

### Immunohistochemistry

Tissues collected from experimental mice were fixed in 10% formalin pH 7.4 (VWR Chemicals) for 24 h before dehydrating in 70% EtOH for 24 h and embedding in paraffin (ThermoScientific, Histostar and Histoplast paraffin). Formalin-fixed and paraffin-embedded lungs and hearts were sectioned onto slides (ThermoScientific Superfrost Ultra-Plus) at 3–5 µm thickness. Immunohistochemistry was performed by de-parafinising (two baths of xylene, for 3 min each), dehydrating (two baths of 100% EtOH, for 3 min each) and rehydrating the slides (in baths of 95, 90, 70 and 50% EtOH for 3 min each). Controlled antigen retrieval was induced with citrate buffer (pH 6.0) for 2 min at 110°C (BioCare Medical, decloaking chamber). Slides were then prepared with Dako EnVision+ Dual link system, HRP kit dual enzyme endogenous blocking buffer for 15 min. Slides were incubated with primary antibody (rabbit anti-mouse phosphorylated H2AX Ser139, Abcam ab11174) diluted 1:500 overnight at 4°C. The secondary detection system was Dako EnVision HRP polymer-labelled rabbit antibodies and DAB was diluted with the substrate buffer 1:25 and washed of the slides with water after 2 min of incubation counterstained with haematoxylin (Sigma-Aldrich) and mounted with coverslips (Sigma-Aldrich DPX mountant for histology, Menzel-Glaser #1.5 coverslips).

## Antibodies

The following antibodies were used for immunoblotting: rabbit polyclonal antisera raised against phosphorylated KAP1 Ser824 (1:1,000, A300-767A, Bethyl Laboratories), KAP1 (1:5,000, A300-274A, Bethyl Laboratories), cleaved PARP1 Asp214 (1:1,000, 9541, Cell Signaling), PARP (1:1,000, 9532, Cell Signalling), ATR (1:2,000, A300-137A, Bethyl Laboratories), CHD4 (1:200, Active Motif, 39289), phosphorylated RPA Ser33 (1:1,000, A300-246A, Bethyl Laboratories), SNM1A (1:1,000, A303-747A, Bethyl Laboratories), FANCD2 (1:2,000, NB100-182, Novus Biologicals) and SMC1 (1:10,000, BL308, Bethyl Laboratories); mouse monoclonal antibodies raised against BRCA2 (1:1,000, OP95, Calbiochem), RPA (1:1,000, ab2175, Abcam), GAPDH (1:30,000, 6C5, Novus Biologicals) and XPF (1:8,000, MS-1381-P0, Thermo Fisher).

## Statistical analysis

Cell line experiments were performed at least three independent times, with technical triplicates for each condition. Results are shown as the average of three independent experiments, unless otherwise indicated, with SEM (standard error of the mean) bars shown for every datapoint. For two-group comparisons data were analysed using unpaired *t*-tests or Mann–Whitney non-parametric test. One-way ANOVA followed by Tukey's *post hoc* test was performed for multiple comparisons. Differences were considered statistically significant if $P < 0.05$.

**Expanded View** for this article is available online.

## Acknowledgements

We are grateful to Dr. Violeta Serra (Vall d'Hebron Institute of Oncology, Barcelona) for providing the VHIO179 PDTX and to Dr. Pavitra Kannan (Department of Oncology, University of Oxford) for her technical help with spheroid cultures. The work in A.B. laboratory was supported by Ministry of Health (Ricerca Corrente 2018) and Italian Association for Cancer Research (IG AIRC grant 21579). This project has received funding from the European Union's Horizon 2020 research and innovation programme under the Marie Skłodowska-Curie grant agreement No. 722729. Research in M.T. laboratory is supported by Cancer Research UK, Medical Research Council and University of Oxford.

## Author contributions

Conception and design: AR, ABi and MT; Acquisition of data (including provided animals, provided facilities, etc.): EMCT, SB, GDG, TR, XL, MP, CF, JM, AK, JBT, DS, MG, SD, JWL, ASB, OMR, ABr, CC, BC, LB, AR, CL, ABi and MT; Writing the manuscript: EMCT and MT; Study supervision: MT.

## Conflict of interest

The authors declare that they have no conflict of interest.

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
