## [Review Process File · EMBO Molecular Medicine]

Chlorambucil targets BRCA1/2-deficient tumours and counteracts PARP inhibitor resistance

Eliana M. C. Tacconi, Sophie Badie, Giuliana De Gregoriis, Timo Reisländer, Xianning Lai, Manuela Porru, Cecilia Folio, John Moore, Arnaud Kopp, Júlia Bagaña Torres, Deborah Sneddon, Marcus Green, Simon Dedic, Jonathan W. Lee, Ankita Sati Batra, Oscar M. Rueda, Alejandra Bruna, Carlo Leonetti, Carlos Caldas, Bart Cornelissen, Laurent Brino, Anderson Ryan, Annamaria Biroccio and Madalena Tarsounas

Review timeline:

Submission date:	26 October 2018
Editorial Decision:	10 December 2018
Revision received:	5 March 2019
Editorial Decision:	8 April 2019
Revision received:	30 April 2019
Accepted:	3 May 2019

Editor: Céline Carret

Transaction Report:

1st Editorial Decision

10 December 2018

Thank you for the submission of your manuscript to EMBO Molecular Medicine. We have now heard back from the two referees whom we asked to evaluate your manuscript. As you will see, both referees are supportive of the work and only offer suggestions to make the paper stronger.

We would therefore welcome the submission of a revised version within three months for further consideration and would like to encourage you to address all the criticisms raised as suggested to improve conclusiveness and clarity. Please note that EMBO Molecular Medicine strongly supports a single round of revision and that, as acceptance or rejection of the manuscript will depend on another round of review, your responses should be as complete as possible.

I look forward to receiving your revised manuscript.

***** Reviewer's comments *****

Referee #1 (Comments on Novelty/Model System for Author):

Medium novelty: The current study identifies the same drugs as potent anticancer agents against homologous recombination deficient cancer cells as Evers et al identified in 2010. Nevertheless, there is added value in the current manuscript as here also BRCA1 deficient cancer cells are interrogated, besides BRCA2-deficient cancer cells. In addition, other BRCA1/2-deficient tumor types besides breast cancer are investigated. Finally, toxicity of cisplatin and chlorambucil are compared in a murine model.

Ethical: It is not clear to this reviewer whether cervical dislocation is an acceptable way of euthanizing mice.

Referee #1 (Remarks for Author):

This interesting study aims to identify drugs that target BRCA1/2-deficient cells and tumours, and compare their activity with well-known drugs like PARP inhibitors and platinum compounds, both in terms of efficacy as well as in terms of toxicity to normal cells and tissues. In addition, the investigators address the question whether some agents may even be active against BRCA1/2-deficient cells that have become resistant to PARP inhibitors or platinum agents. They identify chlorambucil, a drug that has been around since the fifties, as a very potent, and less toxic alternative to platinum agents and PARP inhibitors.

Major
Methods:

Please mention differences between LO-PAC (Evers et al Clin Cancer Res) and Prestwick libraries, to illustrate the added value of the current manuscript.

Results:

1. Figure 1B: DLD1 cells were incubated with either 1.25 μ M olaparib - a very effective dose for this agent - or 0.5 μ M chlorambucil - a dose that yields much less efficacy. Why a suboptimal dose of chlorambucil has been chosen? What happens to DLD1 spheroids +/- BRCA2 with higher doses of chlorambucil?
2. In the in vivo experiment using wild type mice treated with the maximum tolerated doses (MTD) of cisplatin and chlorambucil - for cisplatin MTD a reference has been given. How has the MTD of chlorambucil been determined?
3. Is there anything known about weight loss of the treated mice - per anticancer agent used?

Discussion:

1. Mention something about the risk of induction of secondary malignancies by chlorambucil. Clinicians are always worried about this issue.
2. Is there any evidence from the old literature that especially in young patients chlorambucil, or another alkylating agent, was beneficial in breast or ovarian cancer? Young age is a proxy for a higher chance of being a BRCA1/2 mutation carrier.
3. Mention something about chlorambucil being an oral drug, just like PARP inhibitors.

Minor
There are some typo's throughout the manuscript.

Referee #2 (Remarks for Author):

This manuscript by Tacconi et al. describes a study identifying chlorambucil, a bi-functional alkylating agent, specifically toxic to BRCA1/2-deficient cells, including those resistant to Olaparib and cisplatin. Author demonstrate chlorambucil inflicts replication-associated DNA double-strand

break (DSB)s, similarly to cisplatin, and is substantially less toxic to normal cells and tissues in vitro and in vivo relative to cisplatin.

Cisplatin and Olaparib have demonstrated great effectiveness in treating BRCA-deficient ovarian cancer patients. However, toxic side effects and acquired resistance are still major problems and discovery of drugs that are less toxic while as effective is important. More importantly, chlorambucil is already FDA-approved and currently in clinical use, which may greatly accelerate the process of repurposing chlorambucil for the treatment of BRCA1/2-mutated patients. However there are some major issues that need to be addressed:

1. Authors use 53BP1, Rev7 and Chd4 as models to demonstrate chlorambucil are effective in tumors that are either resistant to cisplatin only or Olaparib only. However, majority of clinical relapsed tumors are resistant to both cisplatin and Olaparib in part because patients on Olaparib treatment have gone through multi-round Cisplatin treatment. Demonstration of effectiveness in model that is resistant to both Cisplatin and Olaparib is needed.
2. Author use MIA PaCa-2 and Capan-1 pair and C4-2 and PEO1 pair to demonstrate chlorambucil specifically target BRCA1 and BRCA2 mutant cells, these cell linea are isogenic cell line which limits the conclusion. There are UWB1.289 and UWB1.289 + BRCA1, PEO1 and PEO4, or induction of BRCA1 loss by CRISPR knock out.
3. In xenografts experiment, chlorambucil treatment show signs of relapse, how does these relapses compare with Cisplatin or Olaparib?
4. In mouse model how does chlorambucil treatment improve PFS when compared with Cisplatin and Olaparib.
5. Does chlorambucil treatment synergize with Cisplatin or Olaparib? Does combination treatment reduce side effect by reducing dose of Cisplatin or Olaparib while maintaining the effectiveness?
6. Figure 6B. To demonstrate higher cisplatin toxicity compared to chlorambucil, demonstration of effectiveness in tumor tissue is needed. At the minimal/low dose (same dose of the drug is not important here.) required to induce DSB in tumor, how much gammaH2AX positive nuclei were scored in normal tissue when treated with chlorambucil versus Cisplatin/Olaparib is more important.
7. BRCA1/2 mutant ovarian tumor often have p53 mutation. It is important to demonstrate p53 independent cell death mechanism.
8. Why in the DLD1 spheroid assay the authors used Olaparib at 1.25uM and Chlorambucil at 0.5uM? Based on the viability assay with the DLD1 cells at 1.25uM Olaparib only ~18% cells survive while at 0.5uM Chlorambucil about 50% of the cells survive. The authors should repeat the experiment with a small range of concentrations up to 1uM that about the 20% of the cells survive.
9. The authors have previously shown that BRCA2-deficient cells are sensitive to MUS81 depletion and that they suffer from replication stress (Lai X et al, 2017). Thus, these cells could be sensitive to ATRi. The authors should compare ATRi with Chlorambucil or even to combine the 2 drugs given that Chlorambucil induces replication stress (Fig 4A).
10. "We found that inactivation of both factors did not further sensitise cells compared to FANCD2 depletion alone (Fig 4E). This demonstrates a concerted action of SNM1A and FANCD2 in response to ICL induction, for the first time placing this nuclease within the FA pathway of ICL repair."

SNM1A depletion has a very minor effect on viability of cells treated with cisplatin (Fig 4E) and the effect with Chlorambucil is only obvious in very high concentrations. The authors could repeat the experiment depleting XPF or ERCC1.

Minor issue:

1. In Figure S1 where PARPi or Cisplatin scored? If they were included in the screen should be indicated.

2. Figure 4A, pRPA blot is not very convincing.
3. Some cell line used are Tp53 deleted or mutated, please label Tp53 status in figure since it is relevant to conclusion.
4. Figure S3C, need to include BRCA1 or BRCA2 knockout/knock down as control.
5. The authors show that Chlorambucil treated cells they do not accumulate in G2/M. Given that these cells have increased levels of DNA damage; the authors should check if the cells are accumulating in S or S/G2 phase.

1st Revision - authors' response

5 March 2019

Referee #1:

Ethical: It is not clear to this reviewer whether cervical dislocation is an acceptable way of euthanizing mice.

Response: According to the latest Laboratory Animal Science Association (LASA) guidelines, cervical dislocation is the preferred method of euthanizing mice, in the hands of an experienced operator. It is one of the methods allowed for small rodents mentioned under Schedule 1 of the UK Animals (Scientific Procedures) Act 1986, the UK legislation regulating animal use for scientific procedures. We mention this legislation in p. 20 of the revised manuscript.

Methods:

Please mention differences between LO-PAC (Evers et al Clin Cancer Res) and Prestwick libraries, to illustrate the added value of the current manuscript.

Response: The LO-PAC library consists of 1,281 drugs and Prestwick of 1,285. The two libraries have 256 drugs (approx. 25%) in common. We have included the LO-PAC library composition, as well as the drugs it has in common with the Prestwick library, in the new Table S1 and referred to it on p. 5 of our new manuscript.

Results:

1. Figure 1B: DLD1 cells were incubated with either 1.25 uM olaparib -a very effective dose for this agent -or 0.5 uM chlorambucil -a dose that yields much less efficacy. Why a suboptimal dose of chlorambucil has been chosen? What happens to DLD1 spheroids +/-BRCA2 with higher doses of chlorambucil?

Response: Chlorambucil doses higher than 0.5 μ M were toxic to both DLD1 $BRCA2^{+/+}$ and $BRCA2^{-/-}$ spheroids.

2. In the in vivo experiment using wild type mice treated with the maximum tolerated doses (MTD) of cisplatin and chlorambucil -for cisplatin MTD a reference has been given. How has the MTD of chlorambucil been determined?

Response: We have added on p. 10 the references: Weisburger *et al*, 1975 and Grosse *et al*, 2009, which indicate 3 mg/kg/day chlorambucil is the MTD in mice.

3. *Is there anything known about weight loss of the treated mice -per anti-cancer agent used?*

Response: We would like to thank the Referee for this suggestion. Indeed, treatment with chlorambucil produced weight loss in 1% of treated mice, compared with cisplatin treatment for which 7% of treated mice showed weight loss. These results are included in the new Table 1 and discussed on p. 11 of the revised manuscript.

Discussion:

1. *Mention something about the risk of induction of secondary malignancies by chlorambucil. Clinicians are always worried about this issue.*

Response: This is an excellent suggestion, which we discussed in a separate paragraph of Discussion (p. 15 of the revised manuscript).

2. *Is there any evidence from the old literature that especially in young patients chlorambucil, or another alkylating agent, was beneficial in breast or ovarian cancer? Young age is a proxy for a higher chance of being a BRCA1/2 mutation carrier.*

Response: We searched for this information through old literature and found no conclusive evidence supporting the idea that young patients may benefit (or not) from treatment with chlorambucil or any other alkylating agents.

3. *Mention something about chlorambucil being an oral drug, just like PARP inhibitors.*

Response: This is an important clinical feature of chlorambucil and we mention it on p. 15 of the revised manuscript.

Referee #2:

1. *Authors use 53BP1, Rev7 and Chd4 as models to demonstrate chlorambucil are effective in tumors that are either resistant to cisplatin only or Olaparib only. However, majority of clinical relapsed tumors are resistant to both cisplatin and Olaparib in part because patients on Olaparib treatment have gone through multi-round Cisplatin treatment. Demonstration of effectiveness in model that is resistant to both Cisplatin and Olaparib is needed.*

Response: To our knowledge, relapsed tumours that are resistant to both cisplatin and olaparib are mostly caused by mutations that reconstitute a wild type allele and BRCA2 expression. The C4-2 human cell line used in Figure 3B was generated through prolonged treatment with cisplatin of the BRCA2-deficient PEO1 cells, as the Referee suggested. This caused reconstitution of BRCA2 expression and resistance to cisplatin and olaparib.

However, clinical cases of olaparib resistance, which retain cisplatin sensitivity are also well-documented. For example, a recent clinical trial (Ang *et al.*, 2013) reported that patients that become resistant to olaparib, having been treated first with cisplatin, continue to respond to cisplatin after they develop olaparib resistance (p. 6 of the revised manuscript). *In vitro* data supporting this conclusion (i.e. that olaparib-resistant BRCA1-defective cells are sensitive to cisplatin) are shown in Fig. 2B and have been also previously published by the group of Andre Nussenzweig (Bunting *et al.*, 2012).

2. *Author use MIA PaCa-2 and Capan-1 pair and C4-2 and PEO1 pair to demonstrate chlorambucil specifically target BRCA1 and BRCA2 mutant cells, these cell line are isogenic cell line which limits the conclusion. There are UWBI.289 and UWBI.289 + BRCA1, PEO1 and PEO4, or inducNon of BRCA1 loss by CRISPR knock out.*

Response: In Fig. 2A we used RPE1 cells in which *BRCA1* deletion was induced with CRISPR/Cas9 system, as the Reviewer requested. These cells are isogenic and therefore our experiments show a clear specific toxicity of chlorambucil against *BRCA1*-deleted cells. Additionally, we use isogenic cell models for *BRCA2* gene deletion in Fig. 1A (DLD1 human cells) and Fig. S5A (HCT116 human cells). Chlorambucil is specifically toxic to the *BRCA2*-deleted cells in both models.

3. In xenografts experiment, chlorambucil treatment show signs of relapse, how does these relapses compare with Cisplatin or Olaparib?

Response: In order to address the Reviewer's question, we performed new xenograft experiments in which we directly compared tumour relapse (as well as other parameters) in *BRCA2*-deficient xenografts treated with chlorambucil, cisplatin and the PARP inhibitor talazoparib. We have included these data in the new Table 1 and described them on p. 11 of the revised manuscript.

4. In mouse model how does chlorambucil treatment improve PFS when compared with Cisplatin and Olaparib.

Response: As in point 3 above, PFS comparison between the three drugs is now included in the new Table 1 and explained on p. 11 of the revised manuscript.

5. Does chlorambucil treatment synergize with Cisplatin or Olaparib? Does combination treatment reduce side effect by reducing dose of Cisplatin or Olaparib while maintaining the effectiveness?

Response: We agree with the Reviewer's view that it would be very interesting to identify drugs that synergise with chlorambucil (i.e. PARP inhibitors, cisplatin, ATR inhibitors and WEE1 inhibitors). This, however, represents an area of investigation beyond the scope of the current study. Synergy studies have just been initiated as a separate research project in the lab and we anticipate that their completion will require a long-term effort.

6. Figure 6B. To demonstrate higher cisplatin toxicity compared to chlorambucil, demonstration of effectiveness in tumor tissue is needed. At the minimal/low dose (same dose of the drug is not important here.) required to induce DSB in tumor, how much gammaH2AX positive nuclei were scored in normal tissue when treated with chlorambucil versus Cisplatin/Olaparib is more important.

Response: We agree with the Reviewer's point that it is necessary to investigate whether chlorambucil and cisplatin can inflict DNA damage not only in healthy tissues (Fig 6), but in tumours as well. In addition to our *in vivo* data shown in Fig 5A, 5D, S5 and S6, we quantified DSB induction in *BRCA2*-deficient xenograft tumours using anti-gH2AX immunohistochemical staining. These data are now included in the new Fig 5D and described on p. 11 of the revised manuscript. We show that chlorambucil and cisplatin increased gH2AX staining in tumours, to a level comparable to the PARP inhibitor talazoparib.

7. *BRCA1/2* mutant ovarian tumor often have *p53* mutation. It is important to demonstrate *p53* independent cell death mechanism.

Response: Indeed, as the Reviewer pointed out, most *BRCA1/2*-mutated ovarian tumours, as well as all cell lines and xenograft tumours used in our study are *p53*-deficient. Understanding the mechanisms of *p53*-independent cell death in these cells and tumours is important. However, this is a broad research field and we anticipate that conclusive results will require a long-term study. Therefore, this area will represent the focus of a separate future research project in the lab.

8. Why in the DLD1 spheroid assay the authors used Olaparib at 1.25 μ M and Chlorambucil at 0.5 μ M? Based on the viability assay with the DLD1 cells at 1.25 μ M Olaparib only ~18% cells survive while at 0.5 μ M Chlorambucil about 50% of the cells survive. The authors should repeat the experiment with a small range of concentrations up to 1 μ M that about the 20% of the cells survive.

Response: Please refer to answer to Referee #1, point 1.

9. The authors have previously shown that BRCA2-deficient cells are sensitive to MUS81 depletion and that they suffer from replication stress (Lai X et al, 2017). Thus, these cells could be sensitive to ATRi. The authors should compare ATRi with Chlorambucil or even to combine the 2 drugs given that Chlorambucil induces replication stress (Fig 4A).

Response: As the Reviewer suggested, we tested the effect of several ATR inhibitors on the viability of BRCA2-defective cells. These inhibitors were found to be highly cytotoxic, regardless of the BRCA2 status, and to have low specificity in targeting BRCA-deficient cells. As an example, we include below data obtained with an ATR inhibitor developed by Bayer, which shows a weak specific effect on the viability of BRCA2-deficient human cells ($n = 3$).

10. "We found that inactivation of both factors did not further sensitise cells compared to FANCD2 depletion alone (Fig 4E). This demonstrates a concerted action of SNM1A and FANCD2 in response to ICL induction, for the first time placing this nuclease within the FA pathway of ICL repair." SNM1A depletion has a very minor effect on viability of cells treated with cisplatin (Fig 4E) and the effect with Chlorambucil is only obvious in very high concentrations. The authors could repeat the experiment depleting XPF or ERCC1.

Response: In response to the Reviewer suggestion, we have depleted XPF using siRNA in human cells and examined their response to chlorambucil and cisplatin treatments. These results are now included in the new Fig S4 and discussed on p. 10 of the revised manuscript.

Minor issue:

1. In Figure S1 where PARPi or Cisplatin scored? If they were included in the screen should be indicated.

Response: The PARP inhibitor olaparib was used as a control in the screen. However, this was not a component of the Prestwick library and thus it is not included in the graph shown in in Fig S1. Similarly, cisplatin was not a component of the Prestwick library.

2. Figure 4A, pRPA blot is not very convincing.

Response: Induction of pRPA in BRCA2-defective DLD1 cells treated with chlorambucil or cisplatin is clearly shown in Fig 4A, as well as in Fig S3A.

3. Some cell line used are Tp53 deleted or mutated, please label Tp53 status in figure since it is relevant to conclusion.

Response: All cell lines used in this work are p53-compromised and we indicate this on p. 16 (Materials and Methods) of the revised manuscript. The exceptions are MRC5 and RPE1 wild type cells used in Fig S3B and Fig S3C, respectively. In response to the Reviewer's suggestion, we indicate the p53-proficient status for these cell lines in the corresponding figures.

4. Figure S3C, need to include BRCA1 or BRCA2 knockout/knock down as control.

Response: BRCA1-deleted RPE-1 cells treated with chlorambucil and cisplatin, which the Reviewer suggested to use as a control, are included in Figure 2A.

5. The authors show that Chlorambucil treated cells they do not accumulate in G2/M. Given that these cells have increased levels of DNA damage; the authors should check if the cells are accumulating in S or S/G2 phase.

Response: In response to the Reviewer's suggestion, we include below the complete FACS profile of the chlorambucil (Chl)-or cisplatin (Cis)-treated DLD1 cells, BRCA2-proficient and -deficient shown in Figure 4C. Cisplatin has the highest impact on G2/M cells, as reported in Figure 4C.

2nd Editorial Decision

8 April 2019

Thank you for the submission of your revised manuscript to EMBO Molecular Medicine. We have now received the enclosed reports from the referees that were asked to re-assess it. As you will see the reviewers are now supportive and I am pleased to inform you that we will be able to accept your manuscript pending minor editorial amendments, which includes text changes commented by referee 1.

I look forward to reading a new revised version of your manuscript as soon as possible.

***** Reviewer's comments *****

Referee #1 (Comments on Novelty/Model System for Author):

The technical quality is high of the revised (and also of the original) manuscript with appropriate controls included in all experiments. The novelty is medium, since similar data have been published before (Evers et al. CCR 2010). Nevertheless, enough novelty is added to the current manuscript that makes its findings also extremely relevant for a subsequent drug re-purposing of chlorambucil in BRCA1/2 deficient tumors.

Referee #1 (Remarks for Author):

The authors have improved the manuscript substantially.

There are just a few things that require attention:

Table 1:

The formula is incorrect: should be something like $(1 - \text{tumor volume treated mice} / \text{tumor volume untreated mice}) \times 100$

If there are 5 mice per experiment, and weight loss is expressed in % of mice with weight loss, then how can this be 1% in an experiment with 5 mice?

Referee #2 (Remarks for Author):

The authors have largely addressed my concerns

2nd Revision - authors' response

30 April 2019

Please address the minor text change commented by referee 1.

Response: In response to the comments of Reviewer 1, we have modified Table 1 by correcting the formula for tumour volume reduction and clarifying the calculation of weight loss in Table legend.

Corresponding Author Name: Madalena Tarsounas and Annamaria Biroccio

Manuscript Number: EMM-2018-09982-V2